

# Individual-level specialisation and interspecific resource partitioning in bees revealed by pollen DNA metabarcoding

Jan Klečka[1], Michael Mikát[2], Pavla Koloušková[1], Jiří Hadrava[1,2] and Jakub Straka[2]

[1] Institute of Entomology, Biology Centre of the Czech Academy of Sciences, České Budějovice, Czech Republic
[2] Department of Zoology, Faculty of Science, Charles University, Prague, Czech Republic

Corresponding author
Jan Klečka, jan.klecka@entu.cas.cz, janklecka.eco@gmail.com

## ABSTRACT

It is increasingly recognised that intraspecific variation in traits, such as morphology, behaviour, or diet is both ubiquitous and ecologically important. While many species of predators and herbivores are known to display high levels of between-individual diet variation, there is a lack of studies on pollinators. It is important to fill in this gap because individual-level specialisation of flower-visiting insects is expected to affect their efficiency as pollinators with consequences for plant reproduction. Accordingly, the aim of our study was to quantify the level of individual-level specialisation and foraging preferences, as well as interspecific resource partitioning, in three co-occurring species of bees of the genus *Ceratina* (Hymenoptera: Apidae: Xylocopinae), *C. chalybea*, *C. nigrolabiata*, and *C. cucurbitina*. We conducted a field experiment where we provided artificial nesting opportunities for the bees and combined a short-term mark-recapture study with the dissection of the bees' nests to obtain repeated samples from individual foraging females and complete pollen provisions from their nests. We used DNA metabarcoding based on the ITS2 locus to identify the composition of the pollen samples. We found that the composition of pollen carried on the bodies of female bees and stored in the brood provisions in their nests significantly differed among the three co-occurring species. At the intraspecific level, individual females consistently differed in their level of specialisation and in the composition of pollen carried on their bodies and stored in their nests. We also demonstrate that higher generalisation at the species level stemmed from larger among-individual variation in diets, as observed in other types of consumers, such as predators. Our study thus reveals how specialisation and foraging preferences of bees change from the scale of individual foraging bouts to complete pollen provisions accumulated in their nests over many days. Such a multi-scale view of foraging behaviour is necessary to improve our understanding of the functioning of plant-flower visitor communities.

## INTRODUCTION

Embracing a multi-level view of foraging specialisation, with the partitioning of individual-level specialisation, between-individual diet variation, and overall population- or species-level specialisation, can shed new light on interspecific interactions (*Brosi, 2016*). Although interspecific differences have traditionally been the main focus of research on resources used by animals (*e.g. Pianka, 1973*; *Abrams, 1980*), intraspecific variation in morphological and physiological traits, behaviour, and diet is common and has important implications for ecological processes at the population and community levels (*Bolnick et al., 2003*; *Bolnick et al., 2011*; *Araújo, Bolnick & Layman, 2011*). For example, between-individual variation in specialisation and dietary preferences may have strong effects on the structure and stability of ecological networks (*Bolnick et al., 2011*). These effects stem from a potential difference between specialisation at the species (or population) level and the individual level. It is important to realise that the total niche width of a species can be decomposed into individual-level niche width and between-individual variation (*Roughgarden, 1972*; *Roughgarden, 1974*). Although the existence of between-individual variation in diet has been recognised for a long time and formed a basis of Van Valen's niche expansion hypothesis (*Van Valen, 1965*), the potential ecological importance of between-individual diet variation has been neglected until a relatively recent resurgence of interest in individual variation (*Bolnick et al., 2003*; *Araújo, Bolnick & Layman, 2011*; *Bolnick et al., 2011*).

So far, it is known that large between-individual variation decreases the strength of intraspecific competition because each individual competes only with a subset of conspecifics, but it may increase the strength of interspecific competition. Species and individuals may thus respond in different ways to changes in the strength of intraspecific or interspecific competition, such as by changing individual diet width (*Fontaine, Collin & Dajoz, 2008*; *Brosi & Briggs, 2013*) or shifting their diets to decrease the level of diet overlap among individuals in the population (*Van Valen, 1965*; *Bolnick et al., 2007*). Different strategies may be employed also by different individuals in the same population. For example, some individuals may be specialised, which makes them more efficient in resource use (*Strickler, 1979*; *Hofstede & Sommeijer, 2006*), while others are more generalised. Switching between resources incurs costs because of memory and learning constraints (*Lewis, 1986*; *Gegear & Laverty, 1998*), but more flexible individuals capable of switching between different types of resources may better cope with spatial and temporal variation in resource availability (*Hofstede & Sommeijer, 2006*). At the species level, there is a strong support for the hypothesis that populations with larger between-individual variation are less vulnerable to environmental changes (*Forsman & Wennersten, 2016*). We thus expect that high between-individual variation in resource use by pollinators may promote the robustness of plant–pollinator networks to habitat destruction or loss of resources in a similar way to foraging flexibility (*Noreika et al., 2019*; *Biella et al., 2019a*; *Biella et al., 2020*).

Substantial evidence demonstrating that many species of animals have high levels of between-individual variation in diets has been accumulated (reviewed in *Bolnick et al., 2003*; *Araújo, Bolnick & Layman, 2011*), but most published studies focused on predators,

particularly vertebrates. We are aware of only two studies on flower-visiting insects which studied the between-individual variation in diets using repeated observations of the same individuals as required to properly describe individual diets (*Bolnick et al., 2002*; *Bolnick et al., 2003*). In their landmark study, *Heinrich (1976)* found that individual bumblebees specialised on different flowering plants not only during a single foraging bout, but also over a longer time frame, although the evidence was rather anecdotal. More recently, *Szigeti et al. (2019)* provided quantitative evidence for between-individual variation in flower visitation by a butterfly, *Parnassius mnemosyne*, partly related to temporal changes in flower abundance. However, more data are needed to test the generality of these results and to evaluate their implications for plant–pollinator interactions (*Brosi, 2016*).

To make matters more complicated, specialisation of an individual may vary at different temporal scales (*Brosi, 2016*). For example, a pollinator may be highly specialised during a single foraging bout, which is often called "floral constancy" or "flower constancy", but it may have a substantially broader diet over its lifetime (*Heinrich, 1976*; *Brosi, 2016*). Flower constancy has been demonstrated in many pollinators, including social and solitary bees, butterflies, and hoverflies (*Heinrich, 1976*; *Waser, 1986*; *Lewis, 1986*; *Goulson & Wright, 1998*; *Slaa, Cevaal & Sommeijer, 1998*; *Amaya-Márquez, 2009*). However, it only considers foraging decisions over a very short temporal scale, often only several consecutive flower visits. We lack information on the variation among foraging bouts of the same individuals over a longer time scale with the few exceptions mentioned above (*Heinrich, 1976*; *Szigeti et al., 2019*).

Studies combining measures of between-individual diet variation and interspecific resource partitioning are needed to shed more light on the community-level consequences of individual-level diet variation. To this end, we studied foraging preferences and specialisation in three sympatric species of bees of the genus *Ceratina*. Specifically, we used pollen DNA metabarcoding to analyse the level of interspecific and intraspecific (among-individual) specialisation of female *Ceratina* bees foraging for pollen. We compared pollen composition among nests build by different females, individual brood cells within the nests, and pollen collected during individual foraging bouts. Our aim was to test whether the three sympatric species differed in their foraging preferences, diet breadth at the species and individual levels, and in between-individual variation in diet composition. Such differences in foraging strategies could decrease the intensity of resource competition and facilitate species coexistence.

## MATERIALS & METHODS

### Species studied

The genus *Ceratina* Latreille, 1802 (Hymenoptera: Apidae: Xylocopinae) is a cosmopolitan genus of bees whose common ancestor was probably facultatively eusocial (*Rehan, Leys & Schwarz, 2012*). Most extant *Ceratina* species are also facultatively eusocial (*Groom & Rehan, 2018*), but the proportion of social nests is generally low, and solitary nesting prevails particularly in temperate climates (*Groom & Rehan, 2018*; *Mikát et al., 2022*). Also, some species are known for complex parental care (*Mikát, Černá & Straka, 2016*), including the

only known example of biparental care in bees (*Mikát et al., 2019*). We focused on three species of the genus *Ceratina*, which are the most abundant bee species at the study site (see below), namely *C. chalybea*, *C. nigrolabiata*, and *C. cucurbitina*. All three species are morphologically similar and live mostly in warm grassland habitats. They build their nests in dead plant stems with soft pith, *e.g.*, of *Rosa canina*, *Centaurea* spp., *Verbascum* spp., *etc.* This makes it easy to study their nesting behaviour and obtain pollen samples from their nests. The nest is made of a linear sequence of brood cell; the innermost brood cell is the eldest (*Rehan & Richards, 2010*). Although biparental care has been observed in *C. nigrolabiata*, only the female provisions the nest, while the male's role is to guard the nest (*Mikát et al., 2019*). Hence, only a single female provisions each nest during the brood establishment in the species we studied. *Ceratina* bees generally perform long-term brood care. After the female finishes brood cell provisioning, she guards the nest and when its offspring reach adulthood, she feeds them by pollen and nectar (*Rehan & Richards, 2010*; *Mikát, Černá & Straka, 2016*; *Mikát et al., 2021*). Although nest abandonment was also observed (*Mikát, Černá & Straka, 2016*; *Mikát, Matoušková & Straka, 2021*), one female usually builds only one nest per life, in contrast to other twig-nesting bees, such as *Hoplitis*, *Hylaeus*, *etc.*, which abandon their nest after the provisioning is finished and therefore are able to establish multiple nests during their lifetime.

Phenology of the three species overlaps and is characterised by two peaks of foraging activity of the nesting females: during brood provisioning and then during the feeding of young adults after emergence (*Mikát, Franchino & Rehan, 2017*). The female guards the nest and is therefore mainly inactive between the periods of brood provisioning and feeding of adult offspring. At our study site, *C. cucurbitina* provisioned its nests mostly in the second half of June and in the beginning of July, and fed its adult offspring in the end of July and in August (*Mikát et al., 2022*). *C. chalybea* provisioned brood cells mainly in the second half of June and fed adult offspring in the second half of July and in August (M. Mikát, 2017, unpublished data). Finally, *C. nigrolabiata* provisioned brood cells in the second half of June and in July and fed adult offspring mainly in August, although it often closes and abandons its nest after the provisioning of brood cells is completed (*Mikát, Matoušková & Straka, 2021*).

## Study site and experimental design

We conducted a field experiment in the Havranické vřesoviště Natural Monument, in the Podyjí National Park, near Znojmo, in the Czech Republic (GPS: 48.8133N, 15.999E) in the spring and summer of 2017. The administration of the Podyjí National Park provided a research permit. The study site and its surroundings comprises a heathland and a dry open grassland with solitary trees and shrubs. We installed artificial nesting opportunities in the grassland following the methods used in previous research at the study site (*Mikát, Černá & Straka, 2016*; *Mikát et al., 2019*). The artificial nesting opportunities consisted of sheaves containing 20 cut stems of *Solidago canadensis*. Each stem was 40 cm long. The sheaves were attached in a vertical orientation to thin bamboo sticks fixed to the ground. We installed ca. 1,300 sheaves as nesting opportunities in April before the beginning of the nesting season. The sheaves were distributed in clusters over an area of ca. 1,500 × 250 m.

We used a small subset of the nests established in the *Solidago* stems for this study, while the remaining nests were used for detailed research on phenology and nesting biology of the three *Ceratina* species (*Mikát, Matoušková & Straka, 2021*; *Mikát et al., 2021*; *Mikát et al., 2022*).

## Sampling in the field

Field sampling consisted of two phases. In the first phase, we collected a subset of the artificial nests (ca. 100 sheaves, each containing 20 *Solidago* stems, spread over an area of ca. 1 ha) on 5 July 2017, *i.e.,* in the end of the nest provisioning period of the three species (see above). We collected occupied nests after the end of the bees' foraging activity, around sunset, when the female bees can be usually found inside the nests (*Mikát, Černá & Straka, 2016*; *Mikát, Franchino & Rehan, 2017*; *Mikát et al., 2019*). We transported the nests to a field laboratory. We then carefully opened the nests with clippers, removed and identified the female to which the nest belonged, and collected the pollen provisions from individual brood cells using sterilised forceps. We stored the samples in individual 2 ml tubes and dried them at room temperature in a desiccator with silica gel. The ID of the nest and the ID of the brood cell within the nest (brood cells ordered as 1, 2, *etc.* starting with the oldest one) was recorded along with information about the developmental stage of offspring in each brood cell (egg, larva with its instar identified, or pupa), which we use to estimate the relative brood cell age. We coded the relative brood cell age as 0 (egg), 1 (1st instar larva), 2 (2nd instar larva), *etc.* We used this relative cell age measure when comparing the composition of pollen samples from individual nest cells to account for the effect of plant phenology (see below)—*e.g.*, cells from two nests containing the same developmental stage were established at a similar date, while a cell containing a second instar larva was older than a cell containing an egg. Most of the nests were not yet fully developed, *i.e.,* they contained mostly eggs and larvae, only some of them contained pupae in the oldest brood cells, and no offspring had matured yet. We collected pollen from brood cells with unconsumed provisions, *i.e.,* those containing eggs or young larvae (alive or dead). In total, we obtained 227 samples from 66 nests of these three species containing a sufficient amount of pollen for the purpose of our analyses (*i.e.,* unconsumed pollen in at least two brood cells); 52 samples from 17 nests of *C. chalybea*, 131 samples from 36 nests of *C. nigrolabiata*, and 44 samples from 13 nests of *C. cucurbitina*. The mean number of brood cells with a sufficient amount of pollen was 3.4 per nest (raw data: https://www.doi.org/10.6084/m9.figshare.13850324).

In the second phase of the fieldwork, we conducted a mark-recapture study of the three *Ceratina* species from 29 July to 1 August 2017, *i.e.,* during the time when the females are mostly feeding adult offspring in their nests. We used 144 artificial nests of the same type as described above arranged individually (not in sheaves) in an array over the area of ca. 10 × 5 m. We individually marked females of the three species captured during the provisioning of their nests. The females were marked by a combination of colour spots on the abdomen. Females were recaptured during four days when they were returning to their nests from foraging bouts. This allowed us to sample pollen collected by the captured female during individual foraging bouts. Capturing the females on return to their nests was facilitated by blocking the entrance to their nests while they were foraging (*Mikát, Franchino & Rehan,*

*2017*). We used sterile aspirators for individual recaptures to prevent contamination. We briefly anaesthetised the captured bee using $CO_2$, scrapped the pollen carried on the underside of the abdomen using a single-use toothpick with a small piece of cotton attached to the end (a miniature analogue of cotton buds for ear cleaning), and stored the pollen in a 2 ml tube. In total, we collected 67 samples; 26 samples from 17 females of *C. chalybea*, 35 samples from 23 females of *C. cucurbitina*, but only six samples from five females of *C. nigrolabiata*. Mean number of captures was 1.5 per female; *i.e.,* most females were captured once or twice (raw data: https://www.doi.org/10.6084/m9.figshare.13850324).

## Pollen DNA metabarcoding

We extracted DNA from the pollen samples using the Macherey-Nagel NucleoSpin Food kit (Macherey-Nagel, Dűren, Germany) according to "the isolation of genomic DNA from honey or pollen" supplementary protocol developed by the manufacturer. Prior to DNA extraction, we homogenised each pollen sample with the CF Buffer from the NucleoSpin Food kit in a 2 ml tube using ceramic beads in a Precellys homogeniser similarly to *Bell et al. (2017)*.

Several genetic markers from the nuclear and chloroplast genome have been used for DNA-based identification of plant material including pollen (*Chen et al., 2010*; *Bell et al., 2016*; *Pornon et al., 2016*; *Kress, 2017*; *Lucas et al., 2018*). We chose to amplify the ITS2 using standard primers for plants (*Chen et al., 2010*), successfully used also in previous studies on pollen metabarcoding (*Sickel et al., 2015*; *Bell et al., 2017*; *Biella et al., 2019a*). Our DNA metabarcoding strategy followed general recommendations by *Taberlet et al. (2018)*. We performed three independent PCR replicates for each sample. The primer design incorporated 8 bp long tags in both the forward and reverse primer, which allowed us to tag individual PCR replicates of individual samples by a unique combination of tags on the forward and reverse primers. The PCR replicates were thus tagged, sequenced together in a single sequencing library, and analysed separately. We used three types of controls: blanks, PCR negative controls, and PCR positive controls. We used a mixture of DNA extracts of five exotic plant species as the PCR positive control. We did the PCR in strips rather than plates to limit cross-contamination (*Kitson et al., 2019*). Each strip contained seven samples and one of the controls. In total, we had 39 blanks, 39 PCR negative controls, and 36 PCR positive controls. The extensive use of different types of controls allowed us to evaluate different sources of contamination and sequencing errors during data analysis (*De Barba et al., 2014*; *Taberlet et al., 2018*). PCR cycles included an initial period of 3 min at 95 °C; followed by 35 cycles of 30 s at 95 °C, 30 s at 55 °C, and 1 min at 72 °C; followed by a final extension of 10 min at 72 °C as in *Bell et al. (2017)*, which seemed to be ideal parameters based also on our preliminary tests. We verified the success of PCR using gel electrophoresis prior to library preparation.

We pooled equal volume of the PCR product from all samples and purified the resulting amplicon pool using magnetic beads (Agencourt AMPure PCR purification kit). The final amplicon pool had a concentration of 52 ng/μl measured by Invitrogen Qubit 3.0 fluorometer (Thermo Fisher Scientific, Waltham, MA, USA). Library preparation was done using a PCR-free approach with Illumina adaptors added by ligation at Fasteris

(Switzerland) and the library was sequenced on Illumina HiSeq 2500 Rapid Run, using 1/10 of the capacity of one sequencing lane resulting in 35,121,401 raw paired reads.

## Reference plant database

We assembled a detailed reference database of ITS2 sequences of most plant species growing in the vicinity of the study site. We attempted to obtain an exhaustive list of plant species growing within the radius of at least 1 km around the study site by our own botanical survey and by extracting data from the literature, particularly a detailed atlas of plants of the Podyjí National Park (*Grulich, 1997*) and a national database of plant records (*Wild et al., 2019*). We collected tissue samples (usually leaves) of most entomogamous plant species we could find in the field and identify reliably and dried them with silica gel. We used the DNEasy Plant Mini Kit (Qiagen, Hilden, Germany) for DNA extraction. We homogenised the leaf samples in a dry state, *i.e.,* without adding the buffer prior to homogenisation. We used the same primers and PCR conditions as for the pollen samples described above. The PCR products were sequenced using Sanger sequencing by Macrogen Europe (Amsterdam, Netherlands).

We complemented our database by ITS2 sequences from GenBank for those plant species we did not sample in the field. We searched for ITS2 sequences of individual plant species and carefully verified the reliability of records for each species to prevent errors from creeping into our reference database. We aligned the sequences in Geneious using the Geneious Aligner and resolved instances of suspected errors on a case by case basis, particularly by checking the sources of the sequences (data from papers by taxonomists were deemed more reliable than data from ecological surveys, samples from geographically close locations were deemed more relevant, *etc.*). Public DNA databases are known to contain numerous misidentified records and other types of errors (*Bridge et al., 2003*). Limiting their impact on our analyses was thus important for confidence in our results.

## Processing of DNA sequencing data

We used the Obitools software (*Boyer et al., 2016*) for bioinformatic processing of the metabarcoding data following general recommendations for filtering and cleaning the sequence data according to *De Barba et al. (2014)* and *Taberlet et al. (2018)*. We first merged the forward and reverse reads and removed low-quality reads with score < 40 or score_norm < 3.9. We then assigned the reads to samples (keeping the three PCR replicates per sample separate) based on the tag sequences and removed reads shorter than 100 bp, based on available data on the length of the ITS2 region in vascular plants (*Chen et al., 2010*). We then dereplicated the reads to obtain the list of unique sequences and their abundance in each sample and PCR replicate. We examined results for the blanks and found that the number of reads in blanks ranged from 0 to 4, so we conservatively discarded all sequences with $<=$ 5 reads for each individual sample/PCR replicate to remove sequencing errors caused by tag jumps (*De Barba et al., 2014*). We then proceeded with sequence identification.

We used our reference database to identify the ITS2 sequences from the samples. We used the *ecotag* function in Obitools to compare each unique ITS2 sequence from the samples

with sequences in the reference database. Sequences were identified at the species or genus level with 0.95 as the minimum sequence similarity threshold for taxonomic assignment. To account for possible incompleteness of our reference database, we examined unidentified sequences and attempted to identify them using BLAST search of the GenBank nucleotide database (https://blast.ncbi.nlm.nih.gov/). We found only a few sequences with low read count not matching data from our reference database which could be identified using BLAST. We updated our reference database with these sequences after verifying that the species concerned are known to occur in the wider area around our study site according to botanical records (*Grulich, 1997*; *Wild et al., 2019*) or could plausibly occur there. We then reran the sequence identification procedure with the updated reference database. The final outcome of species identification was the number of reads per species (or genus) for each sample and PCR replicate.

The next step was a comparison of the three independent PCR replicates for each sample to identify potentially failed or otherwise unreliable PCR replicates (*Taberlet et al., 2018*). We calculated the pairwise overlap of the similarity of the plant species composition for all three combinations of the three PCR replicates for each sample using Pianka's overlap index (*Pianka, 1973*) calculated using the EcoSimR library (*Gotelli, Hart & Ellison, 2015*) in R 4.0.2 (*R Core Team, 2020*). The vast majority of comparisons had overlap >0.99 and the smallest value of an overlap between any two PCR replicates of the same sample was 0.94 in samples from the nests and 0.97 in samples from the bodies of foraging females, indicating that different PCR replicates of the same sample gave highly consistent results in all cases. We then averaged the proportions of reads in the three PCR replicates for each sample for the downstream analyses.

## Data analysis

Statistical analyses of the data were done in R 4.0.2 (*R Core Team, 2020*). We used the Pianka's overlap index (*Pianka, 1973*), an established niche overlap index frequently used to measure diet overlap among potentially competing species of consumers (*Woodward & Hildrew, 2002*; *Klecka & Boukal, 2012*), as a measure of similarity in pollen composition among nests, brood cells, and samples from individual foraging bouts. To test whether there is interspecific resource partitioning among the three *Ceratina* species, we used non-metric multidimensional scaling (NMDS) in two dimensions and ANOSIM—a permutational analysis of similarity (*Clarke, 1993*), both implemented in the vegan library for R (*Oksanen et al., 2019*), using Pianka's overlap index as a measure of similarity in pollen composition among the three species. For the analysis of pollen from the nests, we averaged the proportion of reads per plant species across all brood cells in each nest and we analogously aggregated repeated samples of pollen from the bodies of individual females. These aggregated data were used in NMDS and ANOSIM to test whether the composition of pollen samples differed among the three species.

We also used generalised linear models (GLM) to compare the values of Shannon's $H'$ index calculated from the composition of pollen in individual nests and its components, *i.e.,* the number of plant species and evenness, among the three *Ceratina* species. We included estimated nest age (average relative age of the brood cells based on the

developmental stage—see the section "Sampling in the field" above) as a covariate to account for possible phenological shifts. We used Gaussian error distribution in the analysis of Shannon's $H'$ index and evenness and overdispersed Poisson distribution (quasiPoisson) for the number of plant species. We analogously constructed generalised linear mixed models (GLMM, nest ID used as a factor with random effect) to test whether Shannon's $H'$ index, the number of plant species, and evenness calculated for individual brood cells differed among the three species.

Additional analyses focused on within-individual and between-individual variation in foraging. We used repeatability analysis which partitions within-individual and between-individual variance (*Nakagawa & Schielzeth, 2010*) to evaluate individual-level differences in specialisation and foraging preferences. Specifically, this analysis compared variation in the Shannon's $H'$ index and its components, the number of plant species and evenness, among pollen samples from brood cells from the same nest and among different nests, separately for each of the three *Ceratina* species. We analogously analysed data from pollen samples taken from the bodies of foraging females. We calculated repeatability in rptR library for R (*Schielzeth & Nakagawa, 2013*) using GLMMs fitted by Markov chain Monte Carlo (MCMC) to obtain reliable variance estimates. Response variables were the Shannon's $H'$ index and evenness (GLMMs with Gaussian error distribution) and the number of plant species (a GLMM with Poisson error distribution), individual ID was used as a random effect, and no predictor variable with a fixed effect was included.

We also evaluated individual-level variation in the composition of the pollen samples using two-tailed partial Mantel tests (*Smouse, Long & Sokal, 1986*) implemented in the ecodist library for R (*Goslee & Urban, 2007*). For pollen data from the nests, we constructed a matrix containing the values of the Pianka's overlap index (*Pianka, 1973*) of pollen composition for all pairwise combinations of individual brood cells from all nests of one species. We then transformed the matrix of overlap values into a dissimilarity matrix. Pianka's overlap index varies between 0 and 1, so the dissimilarity of pollen composition between a pair of samples is 1—Pianka's overlap index. A second dissimilarity matrix contained the differences in estimated cell age. Finally, a third dissimilarity matrix contained 0 for all combinations of samples (brood cells) from the same nest and 1 for all combinations of samples from different nests. We then used a two-tailed partial Mantel test (Pearson's correlation, 9,999 permutations) to test whether the dissimilarity of pollen composition among brood cells from the same nest differed from the dissimilarity of pollen composition among brood cells from different nests, conditioned on differences in estimated brood cell age. We did this analysis separately for each of the three *Ceratina* species. We also conducted the same type of analysis using data on the composition of pollen samples obtained from the bodies of foraging females. In this case we tested whether samples repeatedly taken from the same individual are more similar than samples from different individuals.

## RESULTS

Most pollen sequences were identified at the species level, with a few exceptions, *e.g.*, *Rubus* sp. and *Hypericum* sp., where we achieved genus-level identification. Specifically, 90.9%
of reads after quality filtering in samples from the nests of *Ceratina* spp. were identified at the species level, while 9.1% of reads were identified at the genus level and a mere 0.03% remained unidentified. In samples from the bodies of *Ceratina* females, 92.2% of reads after quality filtering were identified at the species level, 7.8% at the genus level and 0.01% were unidentified.

## Interspecific resource partitioning

We found clear interspecific differences in pollen composition in nests from the three *Ceratina* species as well as interspecific differences in their level of specialisation. Overall pollen composition in nests of the three *Ceratina* species, expressed as the mean proportion of reads identified as individual plant species, is summarised in Fig. 1. Nests of *C. chalybea* and the other two species were separated by a NMDS analysis in two dimensions (Fig. 2), while pollen composition in nests of *C. cucurbitina* overlaped with *C. nigrolabiata*. Notably, there was also a much higher spread among individual nests in *C. chalybea* and *C. nigrolabiata* compared to *C. cucurbitina*, see Fig. 2, but this could be partly a consequence of a lower number of observations for *C. cucurbitina*. Differences in pollen composition from nests of the three *Ceratina* species were strongly supported by ANOSIM, a permutational analysis of similarity, using Pianka's overlap index as a measure of similarity in pollen composition ($R = 0.385$, $P < 0.0001$, 9,999 permutations).

Pollen composition in samples collected four weeks later from the bodies of female bees when returning from a foraging bout to the nest shows patterns consistent with data on pollen composition from the nests (Figs. 1 and 2). Samples from *C. chalybea* were again separated from the other two species by a NMDS analysis (Fig. 2). Differences in pollen composition of samples from bodies of the three *Ceratina* species were also strongly supported by ANOSIM ($R = 0.197$, $P = 0.002$, 9999 permutations).

We observed interspecific differences in pollen diversity (Fig. 3) measured by the Shannon's $H'$ index at the level of entire nests (GLM, $F_{2,63} = 3.41$, $P = 0.040$), while the differences at the level of individual brood cells were not statistically significant (GLMM, $\chi^2_2 = 5.01$, $P = 0.082$). Of the two components of the Shannon's $H'$ index, *i.e.*, the number of plant species in a sample and evenness of species composition, only the later differed among the three *Ceratina* species. There was no evidence of interspecific differences in the number of plant species per nest (GLM, $F_{2,63} = 1.35$, $P = 0268$) or in individual brood cells (GLMM, $\chi^2_2 = 0.32$, $P = 0.853$). On the other hand, we found clear differences in evenness among the three species at the level of nests (GLM, $F_{2,63} = 4.54$, $P = 0.015$) as well as brood cells (GLMM, $\chi^2_2 = 8.82$, $P = 0.012$), see Fig. 3. Data on pollen diversity in samples collected from the bodies of female *Ceratina*, *i.e.*, pollen collected during a single foraging bout, showed no significant differences among the three species (Fig. 3).

## Individual-level differences in specialisation and foraging preferences

Females of all three *Ceratina* species showed consistent individual-level differences in their level of specialisation when collecting pollen (Table 1). We found high levels of repeatability of the Shannon's $H$ index of pollen samples in brood cells from individual nests in all three species (median 0.47–0.70), *i.e.*, brood cells in some nests had consistently higher pollen

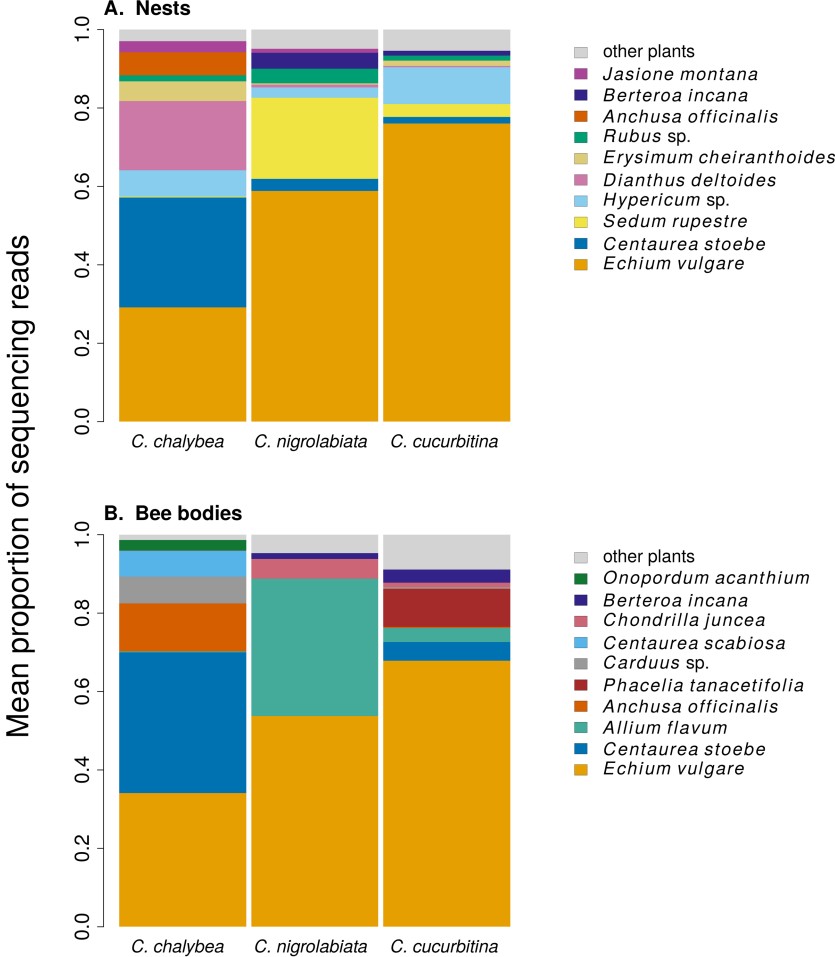

A. Nests

Mean proportion of sequencing reads

other plants
*Jasione montana*
*Berteroa incana*
*Anchusa officinalis*
*Rubus* sp.
*Erysimum cheiranthoides*
*Dianthus deltoides*
*Hypericum* sp.
*Sedum rupestre*
*Centaurea stoebe*
*Echium vulgare*

*C. chalybea*    *C. nigrolabiata*    *C. cucurbitina*

B. Bee bodies

other plants
*Onopordum acanthium*
*Berteroa incana*
*Chondrilla juncea*
*Centaurea scabiosa*
*Carduus* sp.
*Phacelia tanacetifolia*
*Anchusa officinalis*
*Allium flavum*
*Centaurea stoebe*
*Echium vulgare*

*C. chalybea*    *C. nigrolabiata*    *C. cucurbitina*

**Figure 1** **Overall pollen composition of samples from nests and bodies of the three *Ceratina* species.** The composition of pollen samples from the nests (A) and samples collected from the bodies (B) of females of individual species was calculated as the mean proportion of reads assigned to individual plant species. Plant species are sorted from the bottom up according to their total number of reads in samples from all three *Ceratina* species. Ten species with the highest numbers of reads in samples from the nests (A) and females bodies (B) are distinguished by colours, the remaining species are pooled and shown in light grey for clarity.

diversity than brood cells in other nests of the same species. Among the two components of the diversity index (Shannon's *H*), evenness was more strongly repeatable than the number of plant species per brood cell, which had high repeatability only in *C. chalybea* (Table 1).

We also found high repeatability of the Shannon's *H* and evenness in pollen repeatedly sampled from the bodies of individual females returning from a foraging bout (Table 1). We note that 50% of individuals were recaptured on the same day and 50% on multiple different days in *C. chalybea* as well as in *C. cucurbitina*. Only a single *C. nigrolabiata* was recaptured twice, while other individuals were captured only once, so we had to exclude *C. nigrolabiata* from analyses on individual-level differences, because they require repeated sampling of the same individuals.

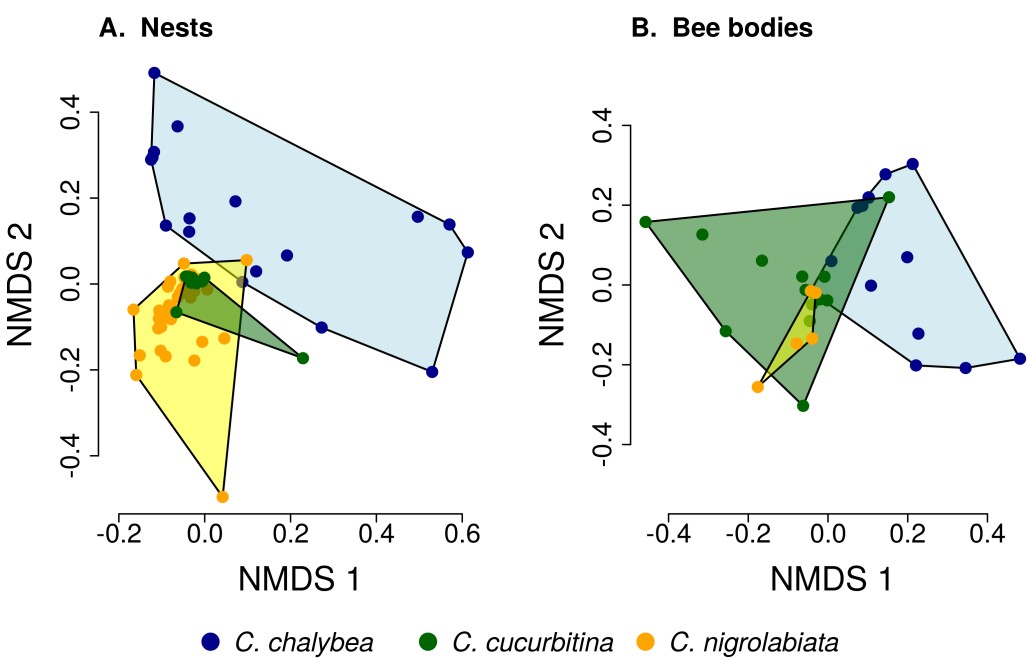

**A. Nests**   **B. Bee bodies**

● *C. chalybea*   ● *C. cucurbitina*   ● *C. nigrolabiata*

**Figure 2** **Similarity of pollen composition in individual nests within and among the three *Ceratina* species.** Results of Non-metric Multidimensional Scaling (NMDS) showing the similarity of pollen composition of samples from individual nests (A) and bodies of individual female bees (B) in two dimensions. The polygons delimit the area containing samples from nests or female bee bodies of individual *Ceratina* species. The position of nests or individual bees is shown by coloured points.

**Table 1** **Repeatability analysis shows consistent differences among individuals in measures of foraging specialisation.** Results of tests of repeatability of pollen diversity in individual females of the three *Ceratina* species are reported. Mean values of repeatability and 95% credible intervals are reported. Values of repeatability significantly larger than zero (shown in bold) mean that variance of Shannon's H', plant species richness, or evenness of pollen composition in brood cells within the same nest (or repeated samples from the body of the same female) was significantly smaller than variance among different nests (or bodies of different females)—this is evidence of consistent among-individual differences. The number of samples from the bodies of *C. nigrolabiata* was not sufficient for analysis.

|  | *C. chalybea* | *C. nigrolabiata* | *C. cucurbitina* |
|---|---|---|---|
| Nests |  |  |  |
| Diversity index (Shannon's *H*) | **0.77** [0.489, 0.878] | **0.45** [0.273, 0.623] | **0.54** [0.280, 0.816] |
| Species richness | **0.53** [0.244, 0.764] | 0.001 [0, 0.171] | 0.28 [0, 0.583] |
| Evenness index (*E*) | **0.36** [0.204, 0.648] | **0.36** [0.203, 0.517] | **0.37** [0.197, 0.700] |
| Bee bodies |  |  |  |
| Diversity index (Shannon's *H*) | **0.46** [0.118, 0.750] | NA | **0.29** [0.058, 0.597] |
| Species richness | 0.002 [0, 0.509] | NA | 0.004 [0, 0.593] |
| Evenness index (*E*) | **0.50** [0.191, 0.767] | NA | **0.64** [0.226, 0.835] |

Individual females of *C. chalybea* and *C. nigrolabiata*, but not *C. cucurbitina*, showed consistent individual-level differences in the composition of pollen contained in brood cells in their nests according to partial Mantel tests conditioned on the temporal distance

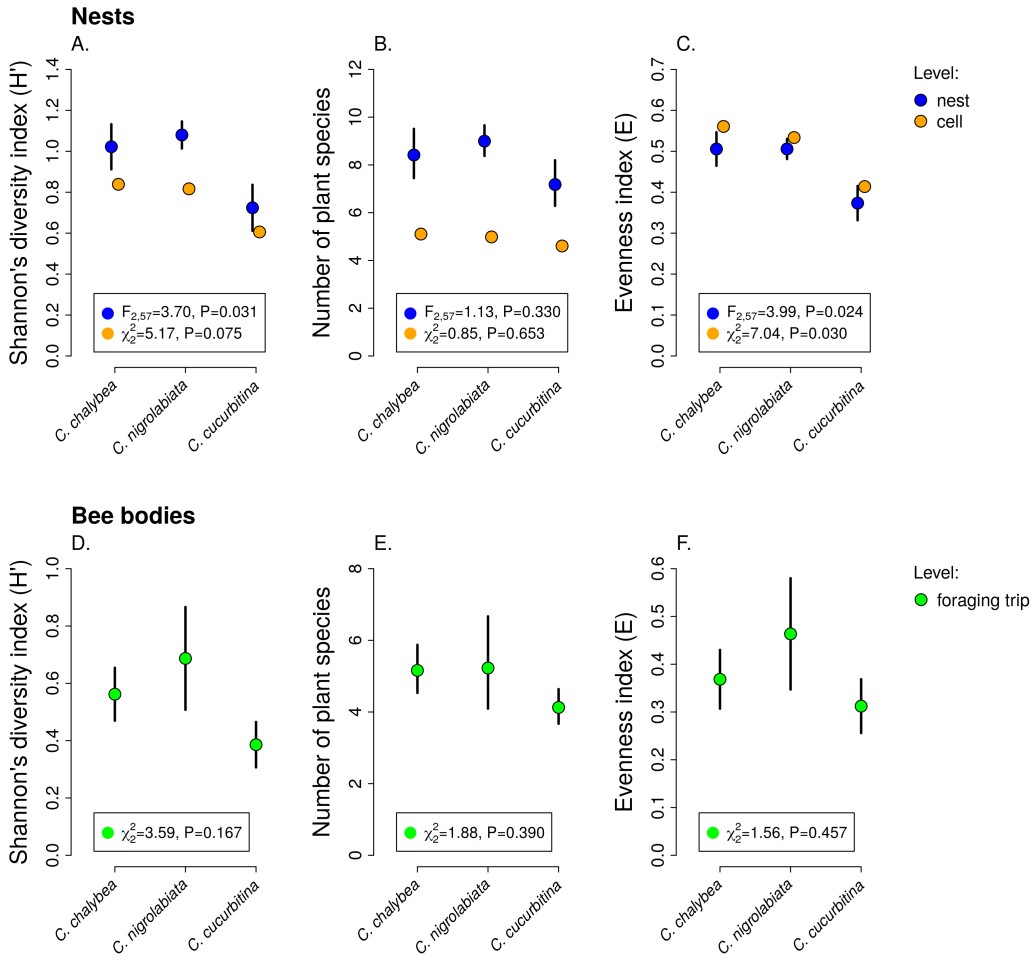

**Figure 3  Pollen diversity of samples at the level of nests, brood cells, and individual foraging bouts.** Shannon's diversity index, plant species richness, and evenness (mean ± SE) calculated from the proportions of reads identified as individual plant species at the level of individual nests and individual brood cells for the three *Ceratina* species (A–C) and for individual foraging bouts (D–F). The *F* and *P* values refer to the results of GLM and the $\chi^2$ and *P* values refer to the results of GLMM (see 'Methods and Results').

of the samples (estimated age of brood cells). Similarity in pollen composition between brood cells from the same nest compared to brood cells from different nests was stronger in *C. chalybea* ($r = 0.135$, $P = 0.0037$) than in *C. nigrolabiata* ($r = 0.052$, $P < 0.0001$), and negligible in *C. cucurbitina* ($r = 0.033$, $P = 0.1764$), based on 9,999 permutations in all cases (Fig. 4). Partial Mantel test accounted for differences in the age of different nests, *i.e.,* changes due to phenology. Indeed, pairs of brood cells with larger differences in their estimated age had more dissimilar pollen composition in all three species (partial Mantel test of the dependence of the dissimilarity of pollen composition on the difference in brood cell age conditioned on whether the pairwise brood cell combination came from the same or different nests, 9,999 permutations): *C. chalybea* ($r = 0.253$, $P = 0.0039$), *C. nigrolabiata* ($r = 0.150$, $P = 0.0003$), and *C. cucurbitina* ($r = 0.415$, $P < 0.0001$).

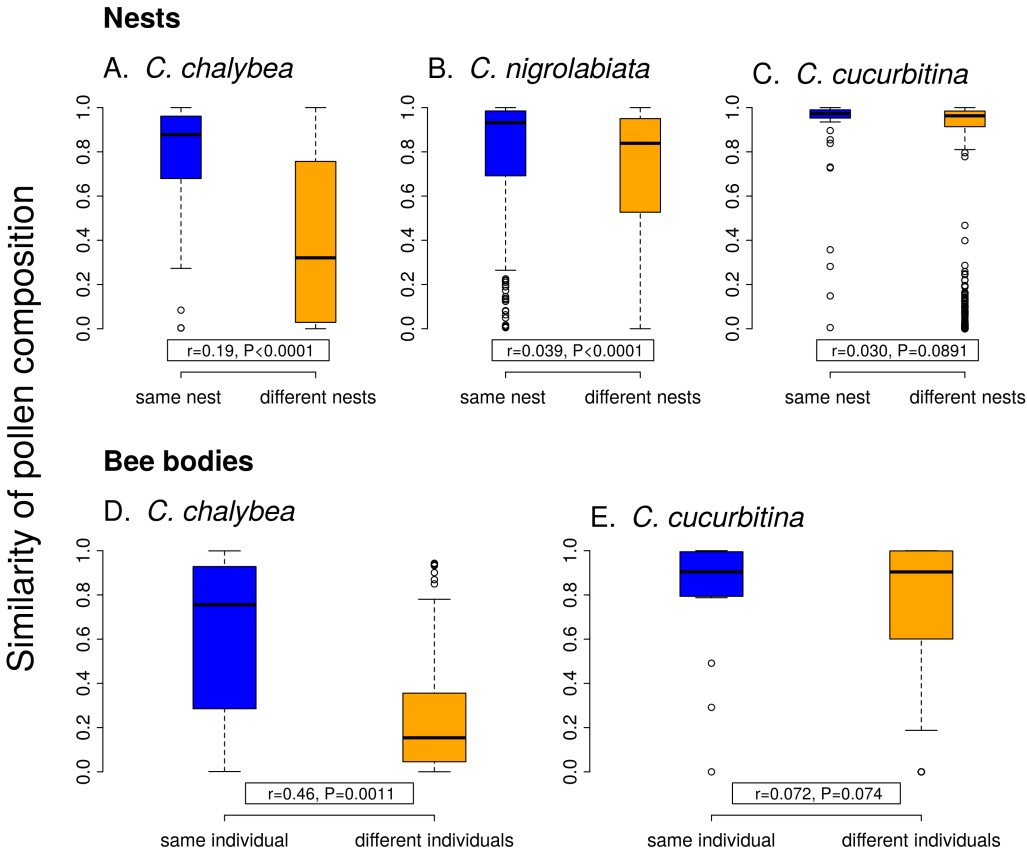

**Figure 4  Within-individual and between-individual variation in pollen composition.** Higher similarity of pollen composition (Pianka's overlap index) between brood cells from the same nest compared to brood cells from different nests (A–C) indicates consistent differences in foraging preferences among individual females. Similarity in pollen composition is analogously compared between repeated samples from the bodies of foraging females (D–E). The number of samples from bodies of *C. nigrolabiata* was not sufficient for analysis. Median and interquartile range is shown in the boxplots. The *r* and *P* values refer to the results of partial Mantel tests (see 'Methods' and 'Results').

Similarly to data from the nests, we found consistent individual-level differences in the composition of pollen sampled directly from bodies of repeatedly captured females of *C. chalybea* returning from a foraging bout (partial Mantel test, $r = 0.460$, $P = 0.0008$, 9,999 permutations), but not in *C. cucurbitina* ($r = 0.072$, $P = 0.1485$). There was no effect of temporal distance (the number of days between collecting the samples) on dissimilarity of pollen composition in both *C. chalybea* ($r = -0.125$, $P = 0.2024$) and *C. cucurbitina* ($r = 0.004$, $P = 0.9799$) (partial Mantel test conditioned on whether pairwise sample combination came from the same or different individual, 9,999 permutations). That means that similarity in the composition of pollen collected from the same individual did not depend on whether the individual was recaptured the same day or several days apart.

## DISCUSSION

### Individual-level consistency and among-individual variation in specialisation and foraging preferences

In this study, we tested how diet breadth and selectivity of three co-occurring species of bees foraging for pollen varies across various levels of aggregation using data from individual foraging bouts and pollen provisions accumulated in the bees' nests. We found that females of all three species displayed consistent among-individual differences in foraging specialisation at the short temporal scale of individual foraging bouts as well as at a longer temporal scale represented by pollen provisions accumulated in brood cells in their nests. Individuals of the more generalised species (*C. chalybea* and to a lesser extent *C. nigrolabiata*) displayed significant among-individual differences in foraging preferences and had larger within-individual variation. On the other hand, individuals of the most specialised species (*C. cucurbitina*) were extremely consistent in their foraging preferences both at the within-individual and among-individual level.

Our data thus support the conceptual scheme of varying levels of specialisation at different levels of aggregation presented by *Brosi (2016)* based on earlier studies on individual-level variation of diet breadth in various consumers (*Bolnick et al., 2003*; *Araújo, Bolnick & Layman, 2011*). As predicted, we found that the three species of solitary bees of the genus *Ceratina* were more specialised at the level of individual foraging bouts than over longer time scales, based on a comparison of pollen diversity in samples from single foraging bouts, pollen provisions in individual brood cells accumulated over a few days, and pollen aggregated in entire nests collected by a single female over the period of many days, see also *Kobayashi-Kidokoro & Higashi (2010)*. Moreover, we found consistent among-individual differences in their specialisation and foraging preferences. Hence, some individuals were consistently more specialised than other individuals of the same species (repeatability analysis; Table 1) and they foraged on a different set of plant species (Fig. 4). Our study thus provides evidence of consistent individual-level specialisation in pollinators following early observations by *Heinrich (1976)* in bumblebees and a recent study on a butterfly (*Szigeti et al., 2019*).

Moreover, we found that larger among-individual differences at the intraspecific level translated into lower specialisation at the species level (Fig. 4). At the species level, *C. chalybea* and *C. nigrolabiata* were more generalised than *C. cucurbitina*. While we observed large differences in pollen composition among different individuals of *C. chalybea*, among-individual differences were smaller in *C. nigrolabiata*, and virtually absent in *C. cucurbitina*, where all individuals were extremely consistent in their specialisation and foraging preferences. Higher generalisation at the species level thus stemmed from larger among-individual variation in diets as observed in other types of consumers, such as predators (*Bolnick et al., 2007*; *Araújo, Bolnick & Layman, 2011*).

As *Bolnick et al. (2003)* and a number of more recent studies on various animals (*Bolnick et al., 2007*; *Araújo, Bolnick & Layman, 2011*; *Lucas et al., 2018*; *Szigeti et al., 2019*; *Lewis, Hughes & Rogers, 2022*) demonstrated, there are two fundamentally different ways to be a dietary generalist. Either all individuals in a population are generalised, or different

individuals specialise on different resources, as we found in *Ceratina* bees. While generalisation measured at the population level may be the same in both cases, differences in the strength of individual-level specialisation may have contrasting implications for population dynamics and for responses of the populations to environmental changes (*Bolnick et al., 2011*). For example, the disappearance of a plant species may be inconsequential for some individual bees, but detrimental for other individual bees of the same species which happen to specialise on it, unless they can adjust their foraging by switching to different plants (*Noreika et al., 2019*; *Biella et al., 2019a*; *Biella et al., 2020*). Another consequence of among-individual differences in resource specialisation of species generalised at the population level but specialised at the individual level is that different individuals in the population experience varying intensity of interspecific competition. This phenomenon may thus have various implications for population biology of generalist species (*Bolnick et al., 2011*).

Interestingly, *C. nigrolabiata* has a longer duration of foraging bouts compared to the other two species (*Mikát et al., 2019*), likely because it has biparental care and the male guards the nest while the female is foraging. Lower time constraints on foraging could promote specialisation on the most rewarding resources (*Lucas, 1983*), but we observed a similar level of specialisation in the three species during single foraging bouts. This may be driven by the same balance of energetic costs and benefits of selective feeding in all three species (*Emlen, 1966*; *Grüter & Ratnieks, 2011*). However, we detected that the three *Ceratina* species differed in their level of within-individual variation over a longer time scale, represented by pollen provisions accumulated in the nests, despite the fact that they were studied at the same site and exposed to the same abundance and composition of resources. These differences may stem from different nutritional demands of the larvae or different levels of foraging flexibility of the three species (*Grüter & Ratnieks, 2011*).

We also conclude that there was a certain level of resource partitioning among the three *Ceratina* species. While pollen from *e.g.*, *Echium vulgare* was found in pollen provisions of all three species, many plant species were found exclusively in pollen provisions of only one or two of the three *Ceratina* species. For example, pollen of *Centaurea stoebe* and *Dianthus deltoides* was common in the samples from *C. chalybea* but almost absent in samples from the other two *Ceratina* species, where it was replaced by pollen of *Sedum rupestre*, *Allium flavum*, and other plants. Such pattern of resource partitioning could be caused by different preferences for floral traits (*Junker et al., 2013*; *Klecka et al., 2018a*), variation in preferred plant height (*Klecka, Hadrava & Koloušková, 2018b*), or by interspecific competition (*Schoener, 1974*; *Palmer, Stanton & Young, 2003*), as demonstrated previously in bumblebees (*Inouye, 1978*; *Graham & Jones, 1996*).

An interesting fact is that *Echium vulgare* was the dominant source of pollen for all three *Ceratina* species, based on the proportion of sequencing reads. *E. vulgare* was among the most abundant flowering plants at the study site and its pollen has a very high protein content (35% crude protein in the dry matter according to *Somerville & Nicol (2006)*), which makes it a potentially excellent resource for bees, but it contains high concentrations of pyrrolizidine alkaloids (*Boppré et al., 2008*; *Lucchetti et al., 2016*; *Trunz et al., 2020*) which are toxic to insects (*Narberhaus, Zintgraf & Dobler, 2005*; *Macel, 2011*). Only a
restricted range of solitary bee species can successfully develop on the pollen of *E. vulgare* (*Praz, Müller & Dorn, 2008*; *Sedivy, Müller & Dorn, 2011*; *Trunz et al., 2020*). In particular, some species of the genus *Hoplitis* (Hymenoptera: Megachilidae) are specialised on *Echium* and other plants in the family Boraginaceae (*Sedivy et al., 2013*), which are also known to contain pyrrolizidine alkaloids (*El-Shazly et al., 1998*; *El-Shazly & Wink, 2014*; *Trunz et al., 2020*). Our results suggest that the three species of *Ceratina* we studied may also have physiological adaptations to develop on the pollen of *E. vulgare*, which allows them to utilise its protein-rich pollen (*Somerville & Nicol, 2006*).

## Implications of variation in specialisation and foraging preferences across individuals and temporal scales

Foraging behaviour of flower visitors has important consequences for the reproduction of entomophilous plants. The reproduction of specialised plants may be negatively affected by fluctuations of the abundance of their pollinators (*Waser et al., 1996*). On the other hand, specialised pollinators may be more effective than generalists by providing higher single visit contribution to plant reproductive fitness (*Larsson, 2005*; *McIntosh, 2005*), although it was reported that specialised solitary bees may remove more pollen per flower visit than generalists, which increases the costs for the plants (*Larsson, 2005*; *Parker, Williams & Thomson, 2016*). However, it is important to emphasise that specialisation specifically at the level of individual foraging bouts, *i.e.,* high flower constancy, matters for pollination because it ensures that pollen is transferred between flowers of the same plant species (*Brosi, 2016*) and it minimises heterospecific pollen transfer which may decrease both the male fitness of the donor plant and the female fitness of the recipient plant (*Waser, 1978*; *Morales & Traveset, 2008*). Hence, even a flower visitor which is generalised at a longer temporal scale (*e.g.*, during it's lifetime), may be a highly efficient pollinator if it temporarily specialises on a single plant species during a foraging bout (*Brosi, 2016*; *Szigeti et al., 2019*). This suggests that pollination efficiency of the three *Ceratina* species we studied may be similar despite their large differences in specialisation at the species level, because they had a comparably high level of specialisation during individual foraging bouts.

Despite the varying level of specialisation, individual brood cells always contained a mixture of pollen of several plant species, which is in line with data on other *Ceratina* species (*Kobayashi-Kidokoro & Higashi, 2010*; *Lawson, Ciaccio & Rehan, 2016*; *McFrederick & Rehan, 2016*). The effect of the composition of pollen provisions on the larval development and survival is not straightforward (*Nicholls & Hempel de Ibarra, 2017*). It has been demonstrated that higher protein content provides benefits for larval development with positive effects persisting to adulthood (*Roulston & Cane, 2002*; *Li et al., 2012*). Accordingly, the most utilised plant by *Ceratina* in our study was *Echium vulgare*, whose pollen is one of the most protein-rich among all plants (*Roulston & Cane, 2000*; *Somerville & Nicol, 2006*). However, pollen from different plants varies widely not only in protein content, but also in energetic value, lipid contents, *etc.* (*Roulston & Cane, 2000*; *Somerville & Nicol, 2006*; *Brodschneider & Crailsheim, 2010*; *Vaudo et al., 2020*). A mixed diet thus may be beneficial for larval development (*Eckhardt et al., 2014*), because it could either better satisfy their nutritional needs or dilute toxins present in some of the food

sources (*Lefcheck et al., 2013*; *Eckhardt et al., 2014*; *Vaudo et al., 2016*). Although we still know little about the importance of resource diversity for the nutrition of solitary bees, it seems that restricted plant diversity caused by climate change or land use change may have a detrimental effect not only on specialised but also on generalised pollinators by affecting their nutrition (*Vaudo et al., 2016*; *Vaudo et al., 2020*), which should be recognised in planning conservation actions (*Vaudo et al., 2015*).

Variation in pollen composition among nests built by different females and even among brood cells in the same nest could lead to differences in the growth and traits of the developing larvae. At the intraspecific level, variation in pollen provisions among nests is an example of maternal effects (*Bernardo, 1996*): the development and traits of the offspring may be driven by individual foraging preferences of their mother. This way, among-individual variation in foraging preferences may promote phenotypic plasticity in the next generation, which may affect evolutionary changes in the solitary bees (*Räsänen & Kruuk, 2007*). At an even finer level, variation in pollen composition among brood cells in the same nest could play an important role in the evolution of eusociality in bees—maternal manipulation of the provisions is known to affect the development of the offspring leading to the production of workers in a facultatively eusocial bee *Megalopta genalis* (Halictidae) (*Kapheim et al., 2011*) and to the production of a dwarf eldest daughter which acts as a worker in *Ceratina calcarata* in North America (*Lawson, Ciaccio & Rehan, 2016*). However, there is no evidence of such maternal manipulation in the three species we studied (*Mikát et al., 2022*; *Mikát et al., 2021*). It is also possible that variation in the composition of the pollen provisions may affect the development of the larvae not only directly by differences in nutritional value, but also indirectly by differences in the composition of bacterial communities in the pollen provisions (*McFrederick & Rehan, 2016*). We are only beginning to understand such implications of individual foraging behaviour, so there is a number of avenues for future research.

## Utility of the DNA metabarcoding approach and limitations of our study

DNA metabarcoding of pollen samples is an efficient method to gain detailed insights into plant–pollinator interactions because it allows us to analyse pollen accumulated by pollinators over multiple flower visits and a single sample thus provides a rich amount of information about flowers visited by an individual pollinator (*Pornon et al., 2017*; *Lucas et al., 2018*; *Pornon et al., 2019*; *Biella et al., 2019b*). Obtaining such detailed insights is facilitated by the use of a rigorous DNA metabarcoding protocol with different types of controls and by a creation of a local reference database which allows us to identify pollen DNA sequences with high level of precision (*Zinger et al., 2019*). In our case, >90% of reads were identified at the species level and almost all the remaining reads at the genus level. This level of precision is unusual when using ITS2 as a marker for plant identification, because many closely related plant species cannot be confidently distinguished. Detailed knowledge of the local flora is thus an important prerequisite for pollen DNA metabarcoding studies where detailed species level data are needed (*Biella et al., 2019b*). We could rely on a long tradition of botanical surveys at the study site and its surroundings (*Grulich, 1997*) to

obtain an exhaustive list of plant species known from the area. However, compiling a database of ITS2 sequences was still complicated by the high frequency of erroneous or spurious records in public databases.

A caveat of using DNA metabarcoding to analyse the composition of pollen samples is that it is not entirely quantitative, *i.e.,* the proportion of reads belonging to a plant species may not be a good proxy for the pollen mass or the number of pollen grains because of different DNA contents per unit mass, amplification bias, *etc.* (*Bell et al., 2016*). However, several studies reported a positive correlation between the number of pollen grains of individual species and the number of sequencing reads using different markers: ITS2 (*Keller et al., 2015*), ITS1, and *trnL* (*Pornon et al., 2016*; *Baksay et al., 2020*), and a positive correlation between observed frequency of flower visits to different plant species and the number of ITS1 or *trnL* reads in pollen loads of pollinators (*Pornon et al., 2016*). These results demonstrate that the proportion of reads may provide at least semi-quantitative information. Importantly, uncertainty in the quantification of the composition of pollen samples is problematic mostly for analyses where absolute quantification is needed, *i.e.,* we have to be careful about making conclusions about the amount of pollen collected by bees based on the number of reads, but it does not invalidate relative comparisons among samples, as done in our study.

Finally, we have to point out that the results described above are based on a short-term study conducted at a single site with a moderate sample size. The first phase of the study used entire nests (17, 36, and 13 per species), each containing on average 3.4 brood cells filled with pollen. This provides a lot of information about the diets of the bees because each brood cell contains pollen accumulated over probably hundreds of foraging bouts. We are thus confident that our data from the nests are robust. On the other hand, the sample size in the second phase of the study (mark-recapture study of foraging females) was suboptimal (17, 23, and five females per species, with a mean of 1.5 captures per females). Foraging activity of the females was lower than expected, probably because of very hot weather, so we collected less samples than we aimed for. We took this into account during data analysis, where we excluded data from *C. nigrolabiata* (only 5 females captured) from those analyses which are more sensitive to sample size (see Table 1 and Fig. 4). In addition, the observed patterns of resource selectivity may be affected by the total abundance and composition of floral resources available at the site at the time of study. Testing how the strength of within-individual and between-individual variation in resource use by pollinators may vary along gradients of plant diversity and resource availability is an interesting avenue for future research.

## CONCLUSIONS

In conclusion, we showed that three species of solitary bees of the genus *Ceratina* were more specialised at the level of individual foraging bouts than over longer time scales. Moreover, we found consistent among-individual differences in their specialisation and foraging preferences. Hence, some individuals were more specialised than other individuals of the same species and differed in the composition of pollen carried on their bodies and stored in

their nests. Our study thus provides evidence of consistent individual-level specialisation in pollinators. Moreover, our data suggest that higher generalisation at the species level may stem from larger among-individual variation in diets as observed in other types of consumers, particularly predators. More detailed knowledge of specialisation and foraging preferences of pollinators across different spatial and temporal scales, from an individual foraging bout to the species level, is necessary to understand plant-flower visitor networks from the functional perspective (*Brosi, 2016*) and to forecast the consequences of various environmental changes on the robustness of plant–pollinator networks. While our study demonstrates the potential importance of individual-level specialisation and interspecific resource partitioning in pollinators, additional studies will be needed to reveal the level of within-individual and between-individual variation in resource use in different types of pollinators and in different ecological contexts.

## ACKNOWLEDGEMENTS

We would like to thank Tereza Hadravová, Jitka Waldhauserová, Marcela Dokulilová, Kateřina Čermáková, Daniela Reiterová, Šimon Zeman, Vojtěch Brož, and Celie Korittová for their help in the field. JK also thanks Pierre Taberlet, Eric Coissac, and their colleagues for sharing their invaluable expertise during the seventh DNA metabarcoding Spring School at Porto, Portugal, in May 2017.

### Funding

This study was funded by the Czech Science Foundation (grants no. GJ17-24795Y and GA20-14872S). The funders had no role in study design, data collection and analysis, decision to publish, or preparation of the manuscript.

### Grant Disclosures

The following grant information was disclosed by the authors:
Czech Science Foundation: GJ17-24795Y, GA20-14872S.

### Competing Interests

The authors declare there are no competing interests.

### Author Contributions

- Jan Klečka conceived and designed the experiments, performed the experiments, analyzed the data, prepared figures and/or tables, authored or reviewed drafts of the article, and approved the final draft.
- Michael Mikát conceived and designed the experiments, performed the experiments, authored or reviewed drafts of the article, and approved the final draft.
- Pavla Koloušková performed the experiments, authored or reviewed drafts of the article, and approved the final draft.
- Jiří Hadrava performed the experiments, authored or reviewed drafts of the article, and approved the final draft.

- Jakub Straka conceived and designed the experiments, authored or reviewed drafts of the article, and approved the final draft.

## Field Study Permissions

The following information was supplied relating to field study approvals (*i.e.*, approving body and any reference numbers):

Field research was permitted by the administration of the Podyjí National Park (permit no. 0188/2017).

## DNA Deposition

The following information was supplied regarding the deposition of DNA sequences:

All sequences are available in GenBank: MZ656004–MZ656078 and MZ661190–MZ661195.

## Data Availability

The data are available in Figshare: Klečka, Jan; Mikát, Michael; Koloušková, Pavla; Hadrava, Jiří; Straka, Jakub (2021): Individual-level specialisation and interspecific resource partitioning in bees revealed by pollen DNA metabarcoding. figshare. Dataset. https://doi.org/10.6084/m9.figshare.13850324.v1.

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
