# Peer review of "Individual-level specialisation and interspecific resource partitioning in bees revealed by pollen DNA metabarcoding"

_PeerJ, doi:10.7717/peerj.13671_

## Round 0.1 · original submission · Major Revisions

Dear authors,

Thank you for your submission. The reviewers have recommended major revisions. Please see their comments below. I look forward to your resubmission.


Reviewer 1 ·

Basic reporting

This study uses state-of-the-art data and methods to further our understanding of how foraging preferences vary within and between closely related bee species. I found this manuscript well written and appropriately structured, besides it follows logically from prior knowledge. In general, the citation of previous work is satisfactory, however, I notice relevant papers regarding the use of DNA metabarcoding to answer similar questions are missing in the list, such as Pornon et al., 2016, 2017 and 2019; and Lucas et al., 2018. The deposited data is accessible, and the figures and tables supporting the findings are clear and readable.

I only have a few concerns regarding some aspects of the abstract and the general background:

L. 20, 31: Please mention what “temporal scales” means here. After reading the entire manuscript, I suggest avoiding this term in the abstract as it can lead to confusion. Please specify in the abstract what you refer to with the temporal scale clearly.
L. 82-84: It is not clear the difference between these two statements; do you mean a change in the diet width vs. a displacement of the diet in the niche axis?
L. 92-93: Please provide a reference for this statement. The two studies cited do not address the effects of between-individual variation on the structure of plant-pollinator networks.
L. 97: Before reading the methods, it is not well understood what “mostly solitary” here means, please reword it.

Experimental design

The research questions are clearly identified and timely; and the motivation for the work is adequately explained. In some cases, more information and details are necessary to improve understanding of the methodology section and for the study to be replicated. My most important concern is related to the insufficient sample size to characterize intraspecific variation in plant resource use and to infer patterns of interspecific resource partitioning. Authors should identify the potential limitations within this research design, which may influence the main outcomes of the study. I believe this study has the potential to advance the field provided these methodological issues can be addressed somehow. More details about this major issue and other minor comments and suggestions:

L. 114: Please provide more phenological characteristics of the three Ceratina species: approximate start and end dates of their activity season, how different are their phenologies?
L. 25-26: Replace “comprises of” with “comprises”.
L. 130: The exact number of sheaves used and the spatial scale (i.e., the size of the study area) should be stated. Also, more details about how these sheaves were spatially distributed (i.e., distance apart) are needed. Could different nesting cavities be used by the same female? How could authors identify whether different nests were constructed by the same or different females?
L. 134: How much time did the sheaves spend in the field? I other words, how much time did the bees have to occupy the artificial nests?
L. 136-137: Only those occupied artificial nest with a female inside were collected? Did you use the artificial nests that were occupied but no female was found inside?
L. 142: How was the relative brood cell age estimated? What relative measure did you use?
L. 144: Replace “has” with “had”.
L. 147-149: I am not sure if this sample size for each bee species is enough to characterize intraspecific variation reliably (only 17, 36 and 13 nests in the case of the first phase of the study; and 17, 23 and 6 females in the second phase). The authors should account for the limitations regarding this small sample size when discussing their results and drawing general conclusions about patterns of intra- and inter-specific variation in resource use.
L. 152: Here the spatial scale is provided, please provide this kind of information also for the first phase of the study.
L. 161-162: I have the same concern regarding the sample size (L. 147-149). Besides, some additional information is needed. How many times was each bee captured during the four days? Please provide a mean value to give an idea of how strong your data on repeated measures of pollen composition from the same female are. It seems that only a few females from each species were recaptured and for most of them you only have one measure.
L. 245-246: Why did you use Pianka’s overlap index? Please provide a justification for using this specific index.
L. 256: How was the average age of the brood cells specified? Which measure did you use?
L. 258: Remove “used”.
L. 259-260: Here and before, it should be clearer what are you testing with each analysis and specifically, which are the response and explanatory variables. In its current form, it makes model construction difficult to understand. For instance, does "to analyse these data at the level of individual brood cells" means "to analyze whether pollen composition differ among brood cells within each nest"?
L. 268-270: What are the predictor variables here?

Validity of the findings

The results are clearly presented, although the clarity of the writing about statistical procedures leading to these results should be improved as stated before. In general, the discussion is well-structured and sound. However, the limitations regarding the insufficient sample size should be stressed out and considered when formulating the conclusions. Based on the results obtained with this sample size, the discussion and main conclusions in their current form may be overreached.

Below I list specific comments and suggestions:

L. 312-313: Maybe this result is due to a low availability of plant resources at that moment?
L. 322-323: How are these results affected by the fact that for most females you collected pollen only once or twice (i.e., you do not have a high number of repeated measures) to evaluate intra-individual differences?
L. 353: I don’t think this study address specifically how resource use varies “through an individual’s lifetime”. Please remove this or focus only on what it is assessed, that is the variation within and between individuals from the same species, and between different species.
L. 362-363: This statement is maybe too speculative and general at this point of the discussion; I suggest removing it and focusing on the main findings here. Nonetheless, I think it is an interesting point to develop later in the discussion.
L. 379-380: To what extent can these differences between the three Ceratina species be attributed to different sample sizes among these species? Besides, more details about the spatial scale of sampling (see comment on L. 130) for the three species are necessary to interpret these results. How were the nests and the individual bees from the different species spatially distributed? Did they show a similar spatial distribution? How could this affect to the intraspecific differences in plant resource use? I mean, I would expect individuals from the same bee species which were flying in distant places to differ more in their resource use than individuals from the same bee species flying in nearby places, just because a matter of local resource availability.
L. 388: Was it tested in this study or only based on previous work?
L. 393-394: I don't think these findings are comparable considering the huge differences in biological characteristics of the two genera, Ceratina and Apis. Please consider removing this example or rewording it to make clear how they are related.
L. 409-412: Could this preference for Echium vulgare be related to a higher relative abundance in the study site?
L. 423: Replace “it’s” with “its”.
L. 423-424: I think it should be indicated here that a high level of specialization in the pollinator assemblage would be also negative from the plant’s perspective, as plant species can be more vulnerable to fluctuations in their pollinator species’ abundance.

·

Basic reporting

I reviewed the manuscript entitled “Individual-level specialization and interspecific resource partitioning in bees revealed by pollen DNA metabarcoding”. In this manuscript the authors deepen into the field of inter and intra-specific variability in the dietary specialization of three bee species of the genus Ceratina.
This is a high-quality research, well-performed and interesting. The overall quality of the writing is very good. My major comment is that in the text sometimes is not clear that the focus is on inter and intra-specific differences, and too much weight is given to the intra-specific level. Is true that this is the main novelty of the article, but more cohesiveness has to be given throughout the text.
I propose a reformulation of the introduction. To put what is now the fourth paragraph as the second, as it is still general. And to give a line to the text that is less focused on the intra-specific differences, to lead the reader to expect that you deal with both, intra- and inter-specific differences in your study. Moreover, the results section is very complex and there are certain results that appear in the discussion and I do not have the impression of having read them in results section, such as the comparison between pollen from bee bodies, cells within nests and nests. May be a table or a figure summarizing all the comparisons you did would help the reader, just conceptual.
The first part of the discussion could be more clear. It is reporting the results and putting them together, but it’s difficult to understand.

Experimental design

Methods in general are very well explained and explained in a high detail, which is great. Still, some points need clarification. The mantel tests need to be better explained. As they are right now is not clear what are the introduced data, and so what is the information extracted from them. And a big part of the article relies on this results.

Validity of the findings

The results of this study are highly relevant, and are supported by the data.

Additional comments

Minor comments:
Please check the format of the references throughout the text (parentheses).
Line 43: “disconnect” -> “disconnection”
Line 46: this line is confusing. The previous sentence is about specialization as a concept, and this is about generalist individuals, but is introduced too roughly.
Line 52: “a large amount of evidence”. Then you should give more references.
Line 66-70. The line about plant fitness is cutting the text. It should me moved elsewhere.
Line 73: remove “on the other hand”.
Line 118: remove “whose relative age is easy to determine, the”. Is redundant with the information given before.
Line 130: “several hundred sheaves’. Could you be more specific?
Line 133. A subset of the occupied nests. Could you be more specific? Just to have an idea.
Lines 213-244. Generate a new subsection in methods called data processing, or data cleaning?
Lines 305-308. Non-significant p-values should be regarded as “no evidence”, not as “little evidence” or “effect to a limited degree”. In the second part of the methods is well-done, no significant results = no effect.
Figure 3. I recommend to modify the legend as to move the legend of the lower row of graphs. I printed the manuscript in black and white and it was challenging to understand the figure from the legend. If the information of the variable employed in each graphs is situated next to them, there will be no confusion.
Lines 357-358. This is very important and is not given sufficient weight in the redaction of the discussion.
Lines 33-385. Please clarify, the sentence is not clear.
Line 423. It’s -> its
Line 470. Number of avenues

---

## Round 0.2 · accepted · Accept

Thank you for submitting your revised manuscript and for addressing the reviewer's comments.

·

Basic reporting

The authors addressed properly all the comments raised in the first round of revision.
The article is coherent, well explained and well performed troughout. It is a pleasure to read and very interesting. Congratulations for all this great work!

One spelling mistake in line 453: "Alium" should be "Allium".

Experimental design

No comment

Validity of the findings

No comment

Additional comments

No comment